# Negative longitudinal magnetoresistance in gallium arsenide quantum wells

Jing Xu[1,2], Meng K. Ma[3], Maksim Sultanov[2], Zhi-Li Xiao [1,2], Yong-Lei Wang[1,4], Dafei Jin[5], Yang-Yang Lyu[1,4], Wei Zhang[1,6], Loren N. Pfeiffer[3], Ken W. West[3], Kirk W. Baldwin[3], Mansour Shayegan[3] & Wai-Kwong Kwok[1]

Negative longitudinal magnetoresistances (NLMRs) have been recently observed in a variety of topological materials and often considered to be associated with Weyl fermions that have a defined chirality. Here we report NLMRs in non-Weyl GaAs quantum wells. In the absence of a magnetic field the quantum wells show a transition from semiconducting-like to metallic behaviour with decreasing temperature. We observe pronounced NLMRs up to 9 Tesla at temperatures above the transition and weak NLMRs in low magnetic fields at temperatures close to the transition and below 5 K. The observed NLMRs show various types of magnetic field behaviour resembling those reported in topological materials. We attribute them to microscopic disorder and use a phenomenological three-resistor model to account for their various features. Our results showcase a contribution of microscopic disorder in the occurrence of unusual phenomena. They may stimulate further work on tuning electronic properties via disorder/defect nano-engineering.

[1] Argonne National Laboratory, Materials Science Division, Argonne, IL 60439, USA. [2] Department of Physics, Northern Illinois University, DeKalb, IL 60115, USA. [3] Department of Electrical Engineering, Princeton University, Princeton, NJ 08544, USA. [4] Research Institute of Superconductor Electronics, School of Electronic Science and Engineering, Nanjing University, 210093 Nanjing, China. [5] Argonne National Laboratory, Center for Nanoscale Materials, Argonne, IL 60439, USA. [6] Department of Physics, Oakland University, Rochester, MI 48309, USA. Correspondence and requests for materials should be addressed to Z.-L.X. (email: xiao@anl.gov) or to Y.-L.W. (email: yongleiwang@nju.edu.cn) or to W.Z. (email: weizhang@oakland.edu)

Magnetic field-induced resistance change is conventionally termed as magnetoresistance (MR), which is usually related to magnetism and plays crucial roles in applications such as sensors and storage devices[1]. In a single-band nonmagnetic material the semiclassical Boltzmann equation approach gives rise to a magnetic field-independent resistivity $\rho_0 \equiv m/e^2 n\tau$, where $e$, $m$, $n$ and $\tau$ are the charge, effective mass, density and relaxation time of the charge carriers[2]. Deviations from a constant resistivity, however, are often reported, for example, in two-dimensional electron gas, where both positive[3] and negative[4] MRs have been observed when the applied magnetic field $\mathbf{B}$ is perpendicular to the current $\mathbf{I}$ ($\mathbf{B} \perp \mathbf{I}$).

With the recent discovery of topological materials, the MR phenomenon has been attracting extensive attention[5–23]. Two of the most remarkable findings are the extremely large MRs for $\mathbf{B} \perp \mathbf{I}$[5–8] and the negative longitudinal MRs (NLMRs) for $\mathbf{B} / / \mathbf{I}$[9–23]. The former can be attributed to the co-existence of high-mobility electrons and holes[5,24], resulting in diminishing Hall effect[24]. The origin of the NLMR, however, is currently under debate. In disordered systems such as films of topological insulator $Bi_2Se_3$[21] and Dirac semimetal $Cd_3As_2$[22], the NLMR is attributed to distorted current paths due to conductivity fluctuations induced by macroscopic disorder as revealed in computer simulations for polycrystalline $Ag_{2\pm x}Se$ samples[25,26]. In contrast, the NLMR in single crystals of these materials[9–17] is often considered as a manifestation of Weyl fermions due to the chirality imbalance in the presence of parallel magnetic and electric fields[27,28]. Recently, theories with topological[29,30] and trivial[30–32] states have been developed to understand the observed NLMR without invoking chiral anomaly. However, some puzzling concerns have been raised on NLMR experiments[33]. For example, numerous non-Weyl materials exhibit NLMRs[20,34–36]. Even within the class of Weyl semimetals, NLMRs are not consistently observed in the same material[37,38]. Furthermore, the reported NLMRs show unpredicted features such as non-monotonic magnetic field behaviour[9,10,16]. More disturbingly, it has been demonstrated that NLMRs can be artificially induced by non-uniform current injection[33,39].

Here we report NLMRs in a non-Weyl material and reveal that microscopic disorder can be their origin. We conduct magneto-transport measurements on GaAs quantum wells, in which microscopic disorder induces unusual phenomena such as quantum Hall plateaus[40] and linear MR (LMR)[3]. We observe NLMRs with both monotonic and non-monotonic magnetic field dependence similar to those reported in topological materials. Furthermore, the NLMRs in our quantum wells exhibit an intriguing temperature behaviour: they occur at temperatures below 5 K, disappear at intermediate temperatures (>5 K) and re-appear at high temperatures ($T > 130$ K and up to 300 K). The NLMR is most pronounced at 160–180 K. After excluding experimental artefacts and accounting for the various features in their magnetic field dependence using a simple three-resistor model, we attribute the observed NLMRs to microscopic disorder including impurity and interface roughness.

## Results

**Negative longitudinal magnetoresistance.** We measured four samples in standard Hall bar geometry, defined through photolithographic patterning. Three of them (see micrographs in Fig.1a and Supplementary Figure 1) are Hall bars on the same wafer and connected to each other, sharing the same applied current and denoted as Samples W1a, W1b and W1c in Supplementary Figure 1. The fourth sample (Sample W2) is the same Hall bar structure fabricated on a separate wafer. All the samples behave

qualitatively the same (see Supplementary Table 1 for a summary of the characteristic parameters including the low-temperature electron densities for all four samples). Here we focus on results of Sample W1b, with additional data from other samples presented in the supplement.

We obtain data on the magnetic field dependence of the sample resistance $R(B)$ at fixed temperatures and angles between the magnetic field and the current (see Fig.1b and Supplementary Figure 9 for the definition of angle $\theta$). At temperatures below 5 K, we observe negative MRs for both $\mathbf{B} / / \mathbf{I}$ ($\theta = 0°$) (Fig. 1e) and $\mathbf{B} \perp \mathbf{I}$ ($\theta = 90°$) (see Supplementary Figure 2). At $T > 5$ K MRs become entirely positive for both magnetic field orientations (see Supplementary Figure 2). With further increase in temperature, the MRs for $\mathbf{B} / / \mathbf{I}$ begin to display non-monotonic behaviour, and NLMR behaviour re-emerges at the medium magnetic fields (see Fig. 1e for MRs at 133 K). At $T > 145$ K and up to room temperature, the MRs become purely negative, although they are non-monotonic at temperatures between 145 and 210 K. From the $R(B)$ curves obtained at various temperatures, we construct the temperature dependence of MR = $[R(B) - R_0]/R_0$, where $R_0$ is the sample resistance in the absence of an external magnetic field. The Cartesian plots of the results for Sample W1b and the other three samples are presented in Fig. 1d and Supplementary Figure 3, respectively. Colour maps that can show both the MR's temperature and magnetic field dependences are given in Supplementary Figure 4 for all four samples. These plots and maps clearly show that all samples exhibit similar NLMRs, with some quantitative variation from sample to sample, for example, the maximum NLMR changes from $-3.85\%$ in Sample W2 at 190 K to $-7.68\%$ in Sample W1c at 165 K (see Supplementary Table 1 for results of other samples). The variations, particularly those in samples patterned on the same wafer, that is, Samples W1a, W1b and W1c, suggest that the NLMR may originate from local properties such as microscopic disorder.

The features including both the monotonic and non-monotonic magnetic field dependences in the $R(B)$ curves shown in Fig. 1e are akin to those reported in crystals[9,10,14,16,17] and films[21,22] of topological materials. The $R(B)$ curves presented in Fig. 1c for Sample W1b at 170 K and various angles indicate that negative MRs only occur when the magnetic field is aligned within a few degrees to the current direction and become most pronounced at $\mathbf{B} / / \mathbf{I}$, akin to those attributed to chiral anomaly[9,10]. However, the temperature dependence of the NLMR in our quantum wells, as shown in Fig.1d for Sample W1b, differs significantly from those reported in the literature, where NLMR diminishes monotonically with increasing temperature[9,10,21,22]. In contrast, the NLMR in our GaAs quantum wells shows a non-monotonic temperature behaviour and becomes most pronounced at ~170 K. This difference could be understood with the existence of two types of microscopic disorders in quantum wells. In addition to the conventional microscopic disorder due to impurity and lattice defects that is relevant at low temperatures, the interface roughness of a quantum well may also induce NLMRs at high temperatures.

Since the samples are in a standard Hall bar geometry defined through photolithographic patterning and the current contacts are far away from the voltage probes, the NLMRs observed here are unlikely to be artefacts arising from non-uniform current injection[33,39]. In general, artefacts become more pronounced with increasing mobility and thus can be excluded with the non-monotonic temperature behaviour of the observed NLMRs, which differs from the monotonic temperature dependence of the mobility (see Fig. 2b and Supplementary Figure 5d). The quantum wells are of very high quality, as demonstrated by the Shubnikov de Haas quantum oscillations at $T = 3$ K (see

Supplementary Figure 6). Their mobility can be larger than $3 \times 10^6 \, \mathrm{cm^2 \, V^{-1} \, s^{-1}}$ at $T = 3 \, \mathrm{K}$ and reach as high as $1.4 \times 10^4 \, \mathrm{cm^2 \, V^{-1} \, s^{-1}}$ even at room temperature (see Supplementary Figure 5d). Therefore, macroscopic defects such as exotic phases and grain boundaries cannot be the origin of NLMRs in the quantum wells.

**Phenomenological three-resistor model.** It is known that microscopic disorder plays a crucial role in the occurrence of quantum Hall plateaus[40], which is observed in all of our quantum wells (see Supplementary Figure 6). That is, areas with microscopic disorder exist in the quantum wells alongside with clean areas (see Fig. 3a for a schematic). Computer simulations for polycrystalline $Ag_{2\pm x}Se$ samples[25,26] have demonstrated that macroscopic disorder such as microscale clusters of excessive Ag induces distorted current paths, resulting in NLMRs. Similarly, the current paths in areas with microscopic disorder are expected to be distorted due to local conductivity fluctuation. Thus, we postulate that the disordered areas where the current paths are distorted in quantum wells have positive (negative) longitudinal magnetoconductances (MRs). Considering that a quadratic field dependence of the magnetoconductance is the most common

relationship revealed by experiments and theories to describe NLMRs, we assume that the magnetotransport behaviour of the disordered areas to be $R_d(B) = R_{d0}/(1 + \alpha B^2)$, where $R_{d0}$ is the resistance of the disordered areas at zero magnetic field and $\alpha$ is a fitting parameter. For the MR of the clean areas where the current paths are not distorted, we take the conventional Drude form $R_c(B) = R_{c0}(1 + \beta B^2)$[27], where $R_{c0}$ is the resistance of the clean areas at zero magnetic field and $\beta$ is a fitting parameter. Figure 3b presents schematics for the expected $R(B)$ relationship for these two scenarios.

The above hypotheses allow us to use a phenomenological three-resistor model to qualitatively account for the various magnetic field behaviours of the observed NLMRs. For a sample with disordered areas mostly surrounded by clean areas (see a schematic in Fig. 3a), its $R(B)$ behaviour can be evaluated using a simplified equivalent circuit consisting of three resistors (see Fig. 3d), that is, a resistor $R_c^p$ with positive MR (representing the clean areas) in parallel with two resistors $R_d$ and $R_c^s$ that are in series and have opposite MRs, where $R_d$ and $R_c^s$ are the resistances of the disordered and clean areas, respectively. The superscripts s and p denote the cases of the clean area in series with and in parallel to the disordered area. In this case the $R(B)$ behaviour can

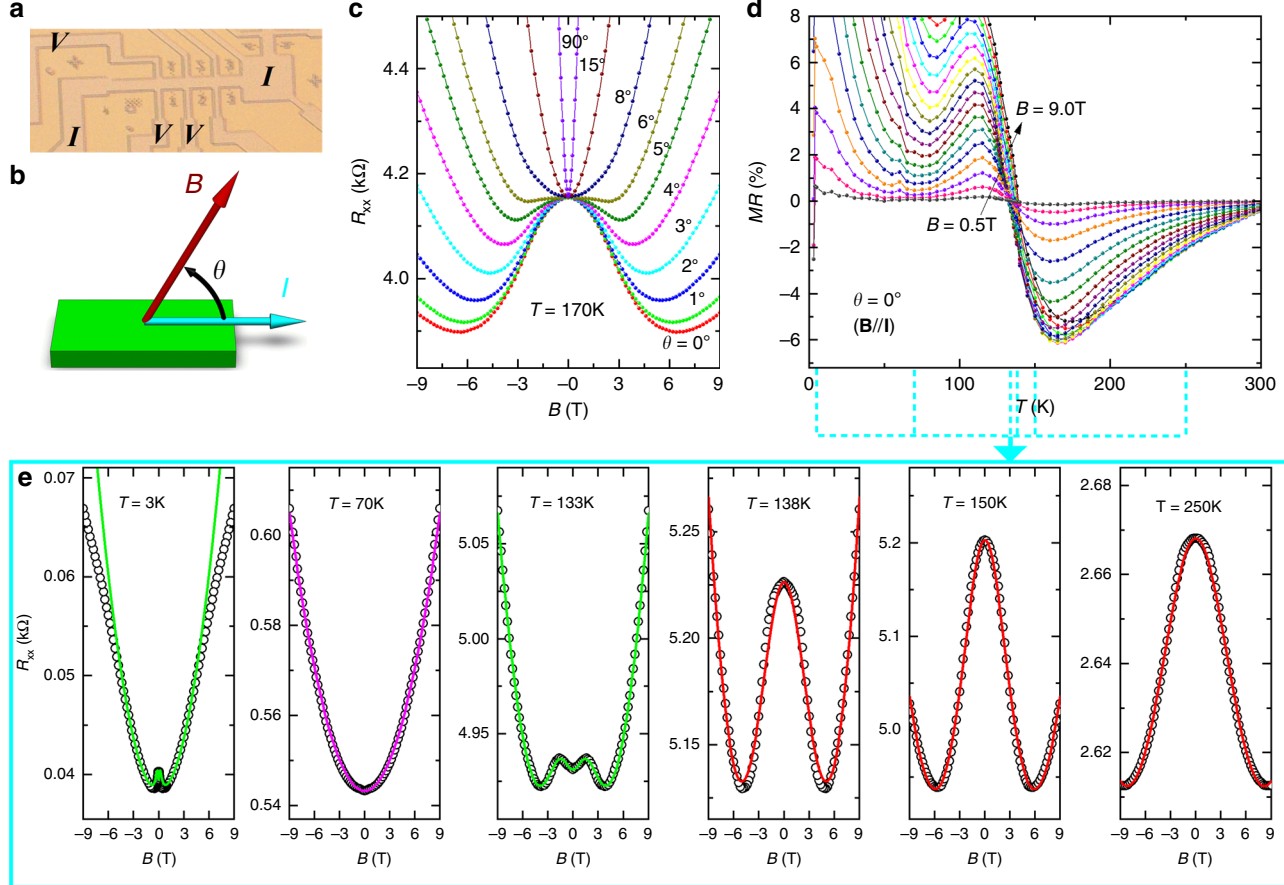

**Fig. 1** Negative longitudinal magnetoresistance in GaAs quantum well. **a** Micrograph of the sample in Hall bar geometry with width of $L_y = 50 \, \mu\mathrm{m}$ and voltage lead distance of $L_x = 100 \, \mu\mathrm{m}$. **b** Schematic showing the definition of the angle $\theta$ between the magnetic field **B** and the direction of the current **I**, with $\theta = 0°$ for **B**//**I** and $\theta = 90°$ for **B**⊥**I**. **c** Magnetic field dependence of the resistance $R_{xx}(B)$ at various field orientations. Negative magnetoresistance can be clearly seen at $\theta \leq 6°$. **d** Temperature dependence of the magnetoresistance (MR) at magnetic fields from $B = 0.5$ to $9.0 \, \mathrm{T}$ at intervals of $0.5 \, \mathrm{T}$ at **B**//**I**, where MR = $[R(B) - R_0]/R_0$, with $R_0$ being the longitudinal resistance $R_{xx}$ at zero field. **e** Representative $R(B)$ curves showing evolution of the MR feature with temperature. The chosen temperatures are given in the corresponding panels and also marked in **d**. In **e**, symbols are experimental data; green lines are fits to the data at $T = 133$ and $3 \, \mathrm{K}$ using Eq. 2 with values of the five variables of $\varepsilon_d = 0.727$, $\gamma_d = 0.295$, $\alpha = 0.148 \, \mathrm{T^{-2}}$, $\beta^s = 6.83 \times 10^{-4} \, \mathrm{T^{-2}}$ and $\beta^p = 0.12 \, \mathrm{T^{-2}}$ and $\varepsilon_d = 0.818$, $\gamma_d = 0.21$, $\alpha = 20 \, \mathrm{T^{-2}}$, $\beta^s = 0.016 \, \mathrm{T^{-2}}$ and $\beta^p = 35 \, \mathrm{T^{-2}}$, respectively; red lines (for $T = 250$, 150 and 138 K) are fits with the reduced form of Eq. 2 for the serial scenario, with fitting parameters presented in Fig. 4, and the magenta line (for $T = 70 \, \mathrm{K}$) describes a quadratic magnetic field dependence $R(B) = R_0 (1 + \beta B^2)$, with $\beta = 1.4 \times 10^{-3} \, \mathrm{T^{-2}}$ and the measured $R_0 = 543.4 \, \Omega$

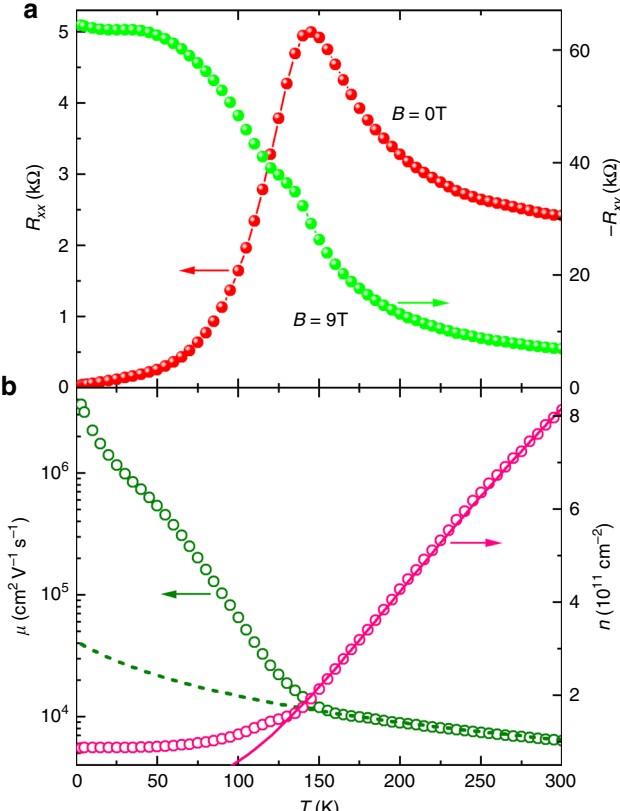

**Fig. 2** Electron density and mobility of GaAs quantum well. **a** Temperature dependence of the zero-field resistance and the Hall resistance $R_{xy}$ at $B = 9$ T and $\theta = 90°$. **b** Corresponding electron density $n$ and mobility $\mu$. The electron densities $n$ in **b** are calculated from the Hall resistances $R_{xy}$ in **a** through the relationship $R_{xy} = B/ne$. The mobilities $\mu$ in **b** are derived from the zero-field resistivity $\rho_0 = R_{xx}(0\,\mathrm{T})L_y/L_x$ in **a** and the electron density $n$ in **b** through the relationship $\rho_0 = 1/ne\mu$. The solid red line in **b** is a fit of $n = N_0 \exp(-E_A/k_B T)$, with $k_B$ the Boltzmann constant, $N_0 = 3.05 \times 10^{12}$ cm$^{-2}$ and $E_A = 34.05$ meV. The dotted olive line in **b** describes a temperature dependence of the electron mobility $\mu = 2.3 \times 10^6/(55 + T)$ (cm$^2$ V$^{-1}$ s$^{-1}$)

be described as

$$R(B) = [1/(R_d + R_c^s) + 1/R_c^p]^{-1}.\qquad(1)$$

Since each of $R_d$, $R_c^s$ and $R_c^p$ has two variables, Eq. 1 has six free parameters. In order to improve the reliability of the analysis, we can take advantage of the measured zero-field resistance by rewriting Eq. 1 as

$$
\begin{aligned}
R(B) = R_0 \big\{ &\varepsilon_d / \big[ \gamma_d / (1 + \alpha B^2) + (1 - \gamma_d)(1 + \beta^s B^2) \big] \\
&+ (1 - \varepsilon_d)/(1 + \beta^p B^2) \big\}^{-1},
\end{aligned}
\qquad(2)
$$

where $R_0 = [1/(R_{d0} + R_{c0}^s) + 1/R_{c0}^p]^{-1}$ is the measured zero-field resistance, $\varepsilon_d = R_{c0}^p/(R_{c0}^p + R_{d0} + R_{c0}^s)$ is the ratio of the conductance of the channel with disordered area to the total value at zero field and $\gamma_d = R_{d0}/(R_{d0} + R_{c0}^s)$ is the ratio of the zero-field resistance of the disordered area to the corresponding total value of the channel consisting of the disordered and clean areas in series.

We use Eq. 2 to account for the experimental $R(B)$ curves as presented in Fig. 1e as solid curves. We find that all five fitting parameters are necessary to describe $R(B)$ curves with positive MR near zero field, like those obtained at $T = 3$ and 133 K,

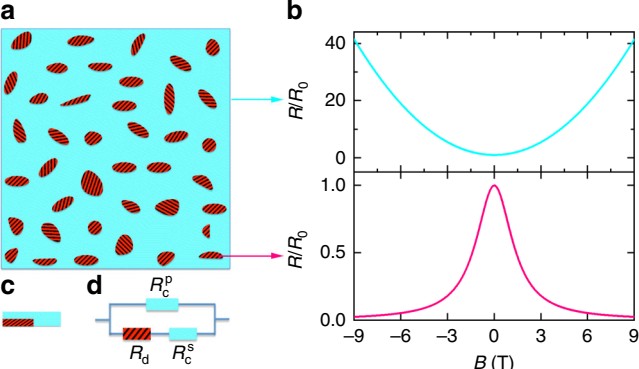

**Fig. 3** Phenomenological three-resistor model. **a** Schematic describing a sample containing areas (red and shaded) where the current paths are distorted and clean areas (cyan) where the current paths are not distorted. **b** Hypothesized magnetic field dependences of the resistance for the clean (cyan line, $R/R_0 = 1 + \alpha B^2$) and disordered regions (pink line, $R/R_0 = 1/(1 + \beta B^2)$, where $\alpha = \beta = 0.5$ T$^{-2}$ are used for the calculations. **c** Simplified picture for electrotransport in the sample. **d** Equivalent circuit for the sample in **c**

resulting in large uncertainty in the analysis. Numerical results in Supplementary Figure 7 indicate that $R_c^p$ is the contributor for the positive MR near zero field (see Supplementary Figure 7c) and the experimental $R(B)$ curves at $T \geq 138$ K in Fig. 1e follow the NLMR behaviour modelled with an equivalent circuit with $R_d$ and $R_c^s$ in series (see Supplementary Figure 7b). In this scenario, $\varepsilon_d = 1$, reducing Eq. 2 to $R(B) = R_0[\gamma_d/(1 + \alpha B^2) + (1 - \gamma_d)(1 + \beta^s B^2)]$. Using the experimentally determined $R_0$, this simplification decreases the number of variables to three ($\gamma_d$, $\alpha$ and $\beta^s$). This simple serial scenario accounts for the experimental data very well, as demonstrated in Fig. 1e for $T = 138$, 150 and 250 K. Figure 4 shows the temperature dependence of the derived $\gamma_d$, $\alpha$ and $\beta^s$. It is reasonable for $\alpha$ and $\beta^s$ to increase with decreasing temperature, since they should reflect the temperature behaviour of the electron mobility (see Fig.2b). The combined effect of $\gamma_d$, $\alpha$ and $\beta^s$ leads to an enhanced NLMR with decreasing temperature. Between 150 and 160 K, however, the NLMRs as well as $\gamma_d$, $\alpha$ and $\beta^s$ change their temperature behaviour, resulting in minima in the NLMR versus temperature curves (see Fig. 1d) and peaks in those of the three fitting parameters.

We note that the change in the temperature behaviour of NLMRs as well as $\gamma_d$, $\alpha$ and $\beta^s$ coincides with a transition (at $T_p \sim 145$ K) from semiconducting-like to metallic temperature dependence in the zero-field $R_0(T)$ curve and a kink in the $R_{xy}$ curve at $B = 9$ T (see Fig. 2a). As shown in Fig. 2b, the temperature dependences of both the electron mobility and density also change significantly at 140–150 K. Quantitatively, the temperature dependence of the electron density for Sample W1b at $T > 140$ K can be well described with $n = N_0 \exp(-E_A/k_B T)$, with $k_B$ the Boltzmann constant, $N_0 = 3.05 \times 10^{12}$ cm$^{-2}$ and $E_A \approx 34$ meV (see Supplementary Table 1 for $N_0$s and $E_A$s of other samples). The $E_A$ value falls in the range of the activation energy of 6–200 meV for Si dopants in the Al$_x$Ga$_{1-x}$As layer ($x = 0$–0.33)[41]. The value of $N_0$ is close to the total Si density of $5.1 \times 10^{12}$ cm$^{-2}$. Thus, the transport at high temperatures is determined by electrons from the Si dopants.

**LMR in perpendicular magnetic fields.** Both experiments[25,26] and simulations[25,26,42] reveal that (quasi-)LMRs at $\mathbf{B} \perp \mathbf{I}$ can appear in polycrystalline materials where NLMRs are detected. As presented in Fig. 5a for sample W1b at $T = 170$ K and various magnetic-field orientations, our quantum wells also exhibit

(quasi-)LMRs at **B⊥I** and NLMRs at **B//I**. Furthermore, the linearity of the MRs for **B⊥I** becomes more pronounced as the NLMRs for **B//I** increase, as demonstrated by the $R(B)$ curves obtained at various temperatures shown in Fig. 5b. The observation of (quasi-)LMR at **B⊥I** further attests to the role of disorder on the magnetotransport of our quantum wells. As demonstrated in Fig. 5c for the $R(B)$ curves obtained at $T = 105$ K, the MR at **B//I** can be positive and even quasi-linear. In this case, (quasi-)LMRs may occur for all field orientations.

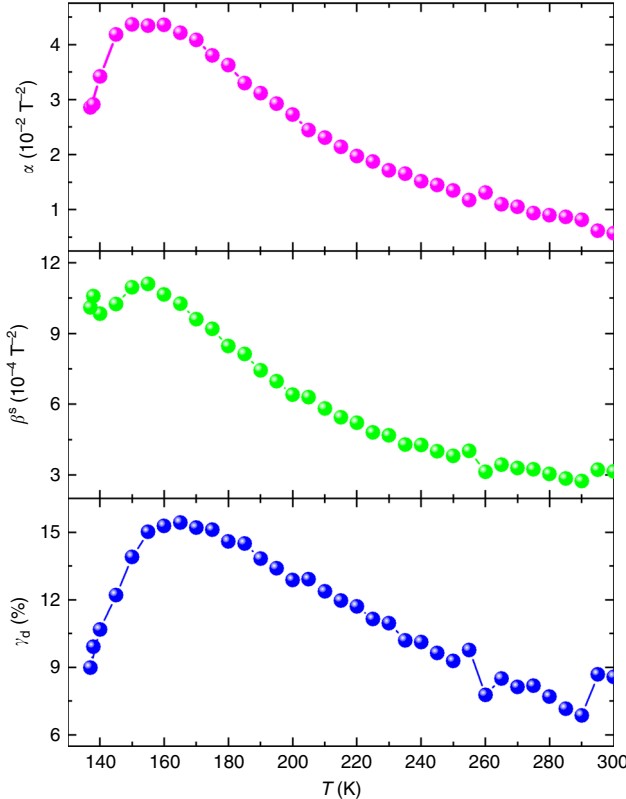

**Fig. 4** Temperature dependence of the derived parameters from the phenomenological model. The experimental longitudinal magnetoresistance $R(B)$ curves at $T \geq 138$ K are fitted with a reduced form of Eq. 2, that is, $R(B) = R_0[\gamma_d/(1 + \alpha B^2) + (1 - \gamma_d)(1 + \beta^s B^2)]$. Using the experimentally determined $R_0$, we need only three variables ($\gamma_d$, $\alpha$, and $\beta^s$) to satisfactorily account for the experimental data, as demonstrated in Fig. 1e for $T = 138$, 150 and 250 K

## Discussion

Our phenomenological three-resistor model and Eq. 2 can be readily applied to account for the NLMRs in films of other materials, in which both micro- and macroscopic disorders are present. The longitudinal magnetoconductance $\sigma_{xx}$ in thin films of topological insulator $Bi_2Se_3$[21] is found to follow $\sigma_{xx} \sim B^2$ in magnetic fields up to 30 T, indicating that the entire sample is disordered. The behaviour of NLMRs in films of Dirac semimetal $Cd_3As_2$ is thickness-dependent[22], with various types of $R(B)$ relationship resembling those in Supplementary Figure 7. In the scenario of Supplementary Figure 7c for a parallel equivalent circuit, Eq. 2 can be re-written as $\sigma(B) = \sigma_{d0}(1 + \alpha B^2) + \sigma_{c0}/(1 + \beta^p B^2)$, which is exactly the same equation used in the analysis of the data for the 370-nm-thick $Cd_3As_2$ film[22]. On the other hand, the two thinner samples (with thicknesses of 85 and 120 nm) exhibit similar type of NLMRs to our quantum wells (of 40 nm thick) at high temperatures, which can be described with the reduced form of Eq. 2 for a series equivalent circuit (see Supplementary Figure 7b). For films of intermediate thicknesses (170 and 340 nm), Eq. 2 with all three resistors (see Supplementary Figure 7a) is required to account for the observed NLMRs.

Recently reported NLMRs in single crystals were often attributed to chiral anomaly. In quantitative analyses[16,34], possible weak anti-localization conductance $\sigma_{WAL}$ and conventional Fermi surface contribution $\sigma_N$ were also included to account for the deviation from quadratic field dependence of the conductance, leading to

$$\sigma(B) = \sigma_{WAL}(1 + C_W B^2) + \sigma_N \qquad (3)$$

with the positive constant $C_W$ reflecting the contribution from chiral anomaly and $\sigma_{WAL} = aB^{1/2} + \sigma_0^{wal}$ with $a < 0$ and $\sigma_N^{-1} = \rho_0^N + bB^2$. Such an expression with five variables could describe the experimental data at low magnetic fields[16,34]. Since weak anti-localization may also exist in quantum wells at low temperatures, we apply Eq. 3 to analyse our data obtained at 3 K and the result is presented in Supplementary Figure 2b. With four fitting parameters ($a = 1.98 \times 10^{-3} T^{-1/2}$, $\sigma_0^{wal} = 2.06 \times 10^{-3}\ \Omega^{-1}$, $C_W = 59.84 T^{-2}$, $b = 87.21\ T^{-2}$) and the measured zero-field resistivity $\rho_0$ ($= 19.81\ \Omega$) (which gives $\sigma_0^N = 1/\rho_0 - \sigma_0^{wal} = 4.84 \times 10^{-2}\ \Omega^{-1}$), Eq. 3 can indeed describe the results at low magnetic fields ($B < 0.5$ T) well and also reproduces the upturn in the MR at high magnetic fields. However, when we set $C_W = 0$ for our non-Weyl quantum well, Eq. 3 reduces to $\sigma(B) = \sigma_{WAL} + \sigma_N$, and yields only positive MR since both $\sigma_{WAL}$ and $\sigma_N$ decrease with increasing magnetic field. Thus, Eq. 3 is not applicable to a non-Weyl system. We also use Eq. 3 to reveal the

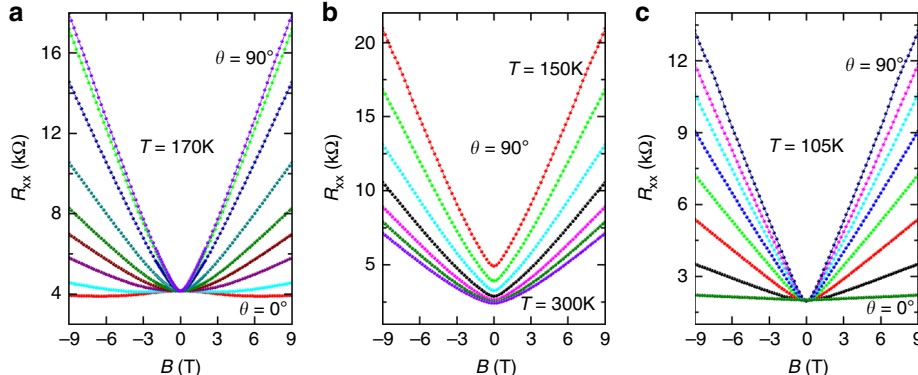

**Fig. 5** Co-existence of linear magnetoresistances (MRs) in perpendicular fields and negative longitudinal MR. **a** $R_{xx}(B)$ curves at $T = 170$ K and various magnetic-field orientations. Quasi-linear MRs can be seen at high angles, while negative longitudinal MR occurs at $\theta = 0°$ (**B//I**). **b** $R_{xx}(B)$ curves at $\theta = 90°$ (**B⊥I**) and temperatures from 150 to 300 K at intervals of 25 K. Linearity of the MRs becomes more pronounced with decreasing temperatures at which negative longitudinal MRs are larger. **c** $R_{xx}(B)$ curves at $T = 105$ K and various angles, showing linear MRs in all magnetic-field orientations

possible contribution of weak localization by allowing $a > 0$. In this case, Eq. 3 cannot describe the experimental data even at low magnetic fields if we set $C_W = 0$. Thus, weak localization that may induce NLMRs in a 2D system at low temperatures[43–45] cannot be the dominant contributor to the observed NLMRs at low temperatures.

Our results reveal that LMRs can co-exist with NLMRs. Thus, NLMRs can be used to distinguish disorder based origin from other mechanisms[46–49] for LMRs. On the other hand, (quasi-) LMRs for $\mathbf{B} \perp \mathbf{I}$ could be an indicator of the existence of NLMRs, providing guidance in the search for materials with NLMRs. As demonstrated in disorder-tuned polycrystalline samples[50], our discovery of the role of microscopic disorder on the occurrence of NLMRs and LMRs will stimulate more work to tune the magnetotransport properties of single crystals through microscopic disorder engineering, for example, doping[16,20] and irradiations with light charge particles such as protons[51] and electrons[52] for crystals.

## Methods

**Sample preparation.** The measured samples are 40-nm-wide GaAs quantum well grown by molecular beam epitaxy[4]. The quantum well is buried 180 nm deep under the surface, and separated by 150-nm $Al_{0.24}Ga_{0.76}As$-thick barriers on both sides from the $\delta$-doped silicon impurities with densities of $1.18 \times 10^{12}$ and $3.92 \times 10^{12}\,cm^{-2}$ for the layers under and above the quantum well, respectively. The samples are fabricated into Hall bars with width of $L_y = 50\,\mu m$ and voltage lead distance of $L_x = 100\,\mu m$ (see Fig. 1a) by photolithography. Contacts to the quantum well are made by annealing InSn at 420 °C for 4 min. The electron densities of the measured quantum wells at 3 K range from 7.91 to 8.83 ($10^{10}\,cm^{-2}$) (see Supplementary Table 1).

**Resistance measurements.** We conduct DC resistance measurements using a Quantum Design Physical Property Measurement System (PPMS-9). We use constant current mode. The results are found to be current-independent (see Supplementary Figure 8). The reported data were taken with $I = 0.5\,\mu A$ for all samples. Angular dependence of the resistance was obtained by placing the sample on a precision, stepper-controlled rotator with an angular resolution of 0.05°. Figure 1b shows the measurement geometry where the angle $\theta$ between the magnetic field $\mathbf{B}$ and the current $\mathbf{I}$ can be varied. Supplementary Figure 9 shows the experimental definition of $\theta = 0°$, that is, the orientation of magnetic field at $\mathbf{B}//\mathbf{I}$. In experiments we measure $R(H)$ curves at various fixed temperatures and angles, as demonstrated in Fig. 1c, e. The MR is defined as $MR = [R(H) - R_0)]/R_0$, where $R(H)$ and $R_0$ are the resistance at a fixed temperature with and without the presence of a magnetic field, respectively.

## Data availability

The data that support the findings of this study are available from the corresponding author upon reasonable request.

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

## Acknowledgements

Magnetotransport measurements were supported by the US Department of Energy, Office of Science, Basic Energy Sciences, Materials Sciences and Engineering. Sample fabrication and characterization were supported by the Department of Energy Basic Energy Sciences (Grant No. DE-FG02-00-ER45841), the National Science Foundation (Grants No. DMR 1709076 and MRSEC DMR 1420541) and the Gordon and Betty Moore Foundation (Grant No. GBMF4420). Use of the Center for Nanoscale Materials, an Office of Science user facility, was supported by the US Department of Energy, Office of Science, Office of Basic Energy Sciences, under Contract No. DE-AC02-06CH11357. W.Z. acknowledges support from the DOE Visiting Faculty Program and the US National Science Foundation under Grants No. DMR-1808892. M.S. was supported by the Fulbright Program. Y.-L.W and Y.-Y.L. acknowledge supports by the National Natural Science Foundation of China (61771235 and 61727805) and the National Key R&D Program of China (2018YFA0209002).

## Author contributions

Z.-L.X., Y.-L.W. and W.Z. designed the experiments; M.K.M., L.N.P., K.W.W., K.W.B., and M.Sh. grew and fabricated the samples. J.X., M.Su, Y.-L.W. and Z.-L.X. conducted the transport measurements, J.X., Y.-L.W., D.J. and W.Z. contributed to data analysis; Z.-L.X., W.Z., Y.-L.W. and W.-K.K. wrote the paper. All of the authors reviewed the manuscript.

## Additional information

**Competing interests:** The authors declare no competing interests.

