## [Peer Review File · Nature Communications]

Reviewers' comments:

Reviewer #1 (Remarks to the Author):

The article reports an observation of negative longitudinal magnetoresistance in GaAs quantum wells. The observed magnetoresistance show dependences resembling those reported in topological materials. This is an important observation which call into question the conventional explanation of the phenomenon in the topological materials based of of chiral anomaly. Therefore I believe that the article should be published.

My only advise to the authors is to make the theoretical part of the article more clear.

In particular, the authors may clarify whether there is an explanation of the theoretical mechanism of the negative longitudinal magnetoresistance which they discuss in the article, or this is purely numerical results. This however should not delay the publication of the article.

Reviewer #2 (Remarks to the Author):

Xu and co-authors report on the observation of a negative longitudinal magneto-resistance in GaAs quantum well. Their research is very interesting and timely. The experiments are carefully performed and the conclusions are solid.

The major issue I have with this paper is the connection and comparison of their data to results in existing literature concerning the observation of negative longitudinal magnetoresistance in topological materials, in particular in Weyl semi-metals. The main difference can actually be seen just looking at figure 1e. As stated by the authors, negative longitudinal magnetoresistance (NLMR) is found above 150 K. In contrast, the NLMR in Weyl semi-metals is more pronounced at lower temperature and becomes weaker with increasing temperature, see e.g. Nature Communications, volume 7, Article number: 11615 (2016). Their simple resistor model is appropriate to describe their observations at high temperature with which they obtain very good fits, however, the underlying scattering mechanisms are related to the underlying systems, i.e. here, a quantum well with a finite well width. Specifically, the population of the second subband in the system seems to be crucial here. The authors mention that inter subband scattering is the dominant scattering mechanism in the temperature range in which NLMR is observed. I do not see how this is relevant, e.g. for Weyl semi-metals. It is actually not the same physics as in Weyl semi-metals but this type of scattering can also lead to NLMR based on entirely different mechanisms in completely different material systems.

I strongly recommend to tone down statements in the abstract and in parts of the introduction that make a direct connection between their 2D system and Weyl semi-metals. The material (systems) under study are completely different, however, a NLMR is observed which is supposed to have a completely different (microscopic) origin. To my opinion, the abstract does not represent the observations reported in the manuscript.

Furthermore, I have one minor question and one minor recommendation.

- (1) I suggest adding self-consistent calculations for the system under investigation.
- (2) Can the authors really refer to their sample as samples with ultra-high mobility?
- (3) Is the expression 're-entrant temperature behavior' justified in the abstract?

To conclude, I emphasize that this manuscript is very relevant and addresses an important field in condensed matter physics, namely, the origin of the NLMR. I would recommend publication of this manuscript after taking into account my recommendations.

Reviewer #3 (Remarks to the Author):

I would like to apologize to the authors for taking longer than usual for providing this report.

I am generally supportive of the manuscript and think that a suitably revised version could appear in Nature Communications.

The experimental results are compelling, and would be a useful addition to the field. However, I find the theoretical analysis lacking and I also do not think the manuscript is well written. I encourage the authors to use the revision process to improve the presentation of results and the flow of ideas.

As mentioned above, the strongest part of the manuscript is the careful experimental data where the authors show the magnetoresistance for GaAs in a Hall bar geometry as a function of the magnetic field angle and temperature.

The authors correctly point out that a subset of the features they observe in this non-topological material, have in the previous literature been attributed to exotic topological effects including the chiral anomaly and the Berry curvature of the electronic bands. Perhaps, if these previous experimental papers had undergone more rigorous review, then such unfounded claims would not have entered the literature. However, given that the negative magnetoresistance is now commonly associated with the chiral anomaly, I am persuaded by the authors for the need for the present paper. This work will provide additional data to the community trying to understand magnetotransport in both topological and non-topological materials.

The other strength of the present manuscript is the discussion ruling out other explanations (e.g. the authors argue that weak localization quantum corrections used by previous analysis can explain some features of their data, but not others), etc. As I write below, this discussion should be expanded and done more clearly.

The weakest part of the paper is the analysis. The authors have a phenomenological equation (Eq 1 and 2) that they use to describe the experiments, plus a qualitative three-resistor model where they put in values by hand, and then get results consistent with Eq. 1 and Eq 2 (and the experimental observations). I would caution against even calling this a theory (as opposed to a qualitative picture). For example, a theory could involve solving the effective medium equations or solving a random resistor network model to capture the scenario shown in Fig. 4a with microscopic inputs, or a microscopic semi-classical Boltzmann calculation (all three of these have been done for the effect of disorder on the linear perpendicular magnetoresistance in other systems, but not for the negative parallel magnetoresistance).

So instead of a theory, the authors present a simple phenomenology that is consistent with their experiment. For the purpose of arguing that in principle, disorder can give rise to the kinds of features seen in the experiment, this is persuasive. The authors have persuaded me that disorder is likely to be the dominant effect, although the exact microscopic mechanism remains elusive. So while it is fine to say that impurities and surface roughness are consistent with the observations, without a theory for either, the language should be more cautious.

I also suggest the following to improve the manuscript:

1. I do not think it is feasible to ask the authors to do a full theory to explain their observations, although that would be nice. But at the very least they should write up exactly what they did (and what parameters they used) as a section of the supplemental material.
2. Also, the longish discussion about a subset of some previous theories [27-30] that do NOT apply to the current experiments seems out of place in the flow of the paper. Maybe part of it could be

moved into the supplementary material. Also, why discuss these theories and not others?

3. At low temperatures weak localization corrections should become important. While I understand that this doesn't explain all the data in the current work, can the authors rule it out completely for the current experiment, especially at low temperature?

4. Some acronyms like XMR are superfluous.

5. Line 148, why write both resistivity and conductivity equations? Anybody can invert one to get the other.

Responses to referee comments and recommendations

We are indebted to all three reviewers for their enthusiastic support in publishing our work in *Nature Communications*. Their insightful and constructive comments are well received and definitely helped to improve the quality of this manuscript. We revised our manuscript according to their recommendations and suggestions, with our point-to-point responses as follows.

Responses to comments and recommendations of Reviewer #1

Reviewer recommendation

The article reports an observation of negative longitudinal magnetoresistance in GaAs quantum wells. The observed magnetoresistances show dependences resembling those reported in topological materials. This is an important observation which call into question the conventional explanation of the phenomenon in the topological materials based of chiral anomaly. Therefore I believe that the article should be published.

Our response

We thank Reviewer 1 for highlighting the importance of our work and for recommending it to be published in *Nature Communications*.

Reviewer comment

My only advise to the authors is to make the theoretical part of the article more clear. In particular, the authors may clarify whether there is an explanation of the theoretical mechanism of the negative longitudinal magnetoresistance which they discuss in the article, or this is purely numerical results. This however should not delay the publication of the article.

Our response

We greatly appreciate Reviewer 1 for this advice, which was also echoed by Reviewer 3 as a weakness of the previous manuscript. In the revised manuscript we moved the numerical derivation of the various characteristic features of NLMR (Fig.3c-e of previous manuscript) to the supplement as Fig.S7 in the revised manuscript, with the caption detailing the analysis procedures and parameters.

As pointed out by Reviewer 3, we do not present a theory on the NLMRs. Instead, we propose a phenomenological three-resistor model to account for the various magnetic field behaviors of the NLMRs.

We significantly revised the text related to the three-resistor model in the manuscript.

Responses to Comments and Recommendations of Reviewer #2

Reviewer comment#1

Xu and co-authors report on the observation of a negative longitudinal magneto-resistance in GaAs quantum well. Their research is very interesting and timely. The experiments are carefully performed and the conclusions are solid.

The major issue I have with this paper is the connection and comparison of their data to results in existing literature concerning the observation of negative longitudinal magnetoresistance in topological

materials, in particular in Weyl semi-metals. The main difference can actually be seen just looking at figure 1e. As stated by the authors, negative longitudinal magnetoresistance (NLMR) is found above 150 K. In contrast, the NLMR in Weyl semi-metals is more pronounced at lower temperature and becomes weaker with increasing temperature, see e.g. Nature Communications, volume 7, Article number: 11615 (2016). Their simple resistor model is appropriate to describe their observations at high temperature with which they obtain very good fits, however, the underlying scattering mechanisms are related to the underlying systems, i.e. here, a quantum well with a finite well width. Specifically, the population of the second subband in the system seems to be crucial here. The authors mention that inter subband scattering is the dominant scattering mechanism in the temperature range in which NLMR is observed. I do not see how this is relevant, e.g. for Weyl semi-metals. It is actually not the same physics as in Weyl semi-metals but this type of scattering can also lead to NLMR based on entirely different mechanisms in completely different material systems.

I strongly recommend to tone down statements in the abstract and in parts of the introduction that make a direct connection between their 2D system and Weyl semi-metals. The material (systems) under study are completely different, however, a NLMR is observed which is supposed to have a completely different (microscopic) origin. To my opinion, the abstract does not represent the observations reported in the manuscript.

Our response

We are delighted that a leading expert says ‘their research is very interesting and timely. The experiments are carefully performed and the conclusions are solid.’

We followed the recommendation of Reviewer 2 to tone down the statements that make a direct connection between our 2D system and Weyl semi-metals. We re-wrote the abstract and re-phrased or removed statements related to Weyl semi-metals in the introduction.

We added the mentioned article [*Nat. Commun.* **7**, 11615 (2016)] to the references (Ref.18 in the revised manuscript)

Reviewer comment#2

Furthermore, I have one minor question and one minor recommendation.

- (1) I suggest adding self-consistent calculations for the system under investigation.
- (2) Can the authors really refer to their sample as samples with ultra-high mobility?
- (3) Is the expression ‘re-entrant temperature behavior’ justified in the abstract?

Our response

- (1) In the manuscript we present experimental observation and a phenomenological model. We hope that our model can stimulate more rigorous theoretical work, which is currently beyond our expertise.
- (2) The mobilities (absolute values and temperature dependence) of our quantum wells are comparable to those in Ref.3 of this manuscript, which uses ‘ultra-high mobility’ to describe the quantum well in its title. In order to avoid confusion, however, we removed ‘ultra-high’ from the revised manuscript.
- (3) We removed the term of ‘re-entrant’ in the revised manuscript.

Reviewer recommendation

To conclude, I emphasize that this manuscript is very relevant and addresses an important field in condensed matter physics, namely, the origin of the NLMR. I would recommend publication of this manuscript after taking into account my recommendations.

Our response

We thank Reviewer 2 for his/her positive judgement on our work and recommendation on publishing it in *Nature Communications*. As stated in our responses to his/her comments 1 and 2, we seriously took his/her insightful recommendations and revised the manuscript accordingly.

Responses to Comments and Recommendations of Reviewer #3

Reviewer recommendations

I would like to apologize to the authors for taking longer than usual for providing this report. I am generally supportive of the manuscript and think that a suitably revised version could appear in Nature Communications.

The experimental results are compelling, and would be a useful addition to the field. However, I find the theoretical analysis lacking and I also do not think the manuscript is well written. I encourage the authors to use the revision process to improve the presentation of results and the flow of ideas.

Our response

We thank Reviewer 3 for carefully reading the manuscript and for the constructive suggestions for improving the presentation of our results and the flow of ideas. We revised the manuscript according to his/her comment#2 below.

Reviewer comment#1

As mentioned above, the strongest part of the manuscript is the careful experimental data where the authors show the magnetoresistance for GaAs in a Hall bar geometry as a function of the magnetic field angle and temperature.

The authors correctly point out that a subset of the features they observe in this non-topological material, have in the previous literature been attributed to exotic topological effects including the chiral anomaly and the Berry curvature of the electronic bands. Perhaps, if these previous experimental papers had undergone more rigorous review, then such unfounded claims would not have entered the literature. However, given that the negative magnetoresistance is now commonly associated with the chiral anomaly, I am persuaded by the authors for the need for the present paper. This work will provide additional data to the community trying to understand magnetotransport in both topological and non-topological materials.

The other strength of the present manuscript is the discussion ruling out other explanations (e.g. the authors argue that weak localization quantum corrections used by previous analysis can explain some features of their data, but not others), etc. As I write below, this discussion should be expanded and done more clearly.

The weakest part of the paper is the analysis. The authors have a phenomenological equation (Eq 1 and 2) that they use to describe the experiments, plus a qualitative three-resistor model where they put in values by hand, and then get results consistent with Eq. 1 and Eq 2 (and the experimental observations). I would caution against even calling this a theory (as opposed to a qualitative picture). For example, a theory could involve solving the effective medium equations or solving a random resistor network model to capture the scenario shown in Fig. 4a with microscopic inputs, or a microscopic semi-classical Boltzmann calculation (all three of these have been done for the effect of disorder on the linear perpendicular magnetoresistance in other systems, but not for the negative parallel magnetoresistance).

So instead of a theory, the authors present a simple phenomenology that is consistent with their experiment. For the purpose of arguing that in principle, disorder can give rise to the kinds of features seen in the experiment, this is persuasive. The authors have persuaded me that disorder is likely to be the dominant effect, although the exact microscopic mechanism remains elusive. So while it is fine to say that impurities and surface roughness are consistent with the observations, without a theory for either, the language should be more cautious.

Our response

Yes, we present a simple phenomenological model rather than a theory to account for the observed NLMRs. We hope our work will stimulate more rigorous theoretical investigations and/or computer simulations on NLMRs in disordered systems. In this regard, we added the work by Parish and Littlewood [*Nature* **426**, 162 (2003)] to the references (Ref.42). In order to explain the non-saturating (linear) magnetoresistance for $B \perp I$, they modeled a strongly inhomogeneous conductor problem by discretization into a random resistor network that they analyzed numerically.

We agree with the reviewer on his/her judgement on the strengths and weakness of our previous manuscript. We revised the analysis part significantly (please see our response to the items #1-3 in his/her comments #2)

Reviewer comment#2

I also suggest the following to improve the manuscript:

- 1. I do not think it is feasible to ask the authors to do a full theory to explain their observations, although that would be nice. But at the very least they should write up exactly what they did (and what parameters they used) as a section of the supplemental material.*
- 2. Also, the longish discussion about a subset of some previous theories [27-30] that do NOT apply to the current experiments seems out of place in the flow of the paper. Maybe part of it could be moved into the supplementary material. Also, why discuss these theories and not others?*
- 3. At low temperatures weak localization corrections should become important. While I understand that this doesn't explain all the data in the current work, can the authors rule it out completely for the current experiment, especially at low temperature?*
- 4. Some acronyms like XMR are superfluous.*
- 5. Line 148, why write both resistivity and conductivity equations? Anybody can invert one to get the other.*

Our response

We greatly appreciate Reviewer 3 for these valuable suggestions on the presentation of this work, which definitely help to improve the quality of this manuscript. Following the reviewer's advices, we made the following changes:

1. We separated Fig.3 in the previous version into Fig.3 and Fig.S7 in the revised manuscript. We present a detailed explanation of the numerical calculations and their parameters in the caption of Fig.S7. The parameters derived using Eq.2 to fit the experimental data are presented in the caption of Fig.1 (for $T = 3$ K and 133K) and in Fig.4 (for $T \geq 138$ K).
2. We deleted the discussion (and the associated Fig.S7 in previous manuscript) about a subset of some previous theories [27-30], which were developed recently to explain the NLMRs without invoking chiral anomaly.

3. We expanded the discussion on possible explanation of NLMRs at low temperatures (data at $T = 3$ K) with weak localization or weak anti-localization corrections on pages 9 and 10 and a fitting curve with Eq.3 was added to Fig.S2b.
4. Superfluous acronyms XMR and 2DEG have been removed in the revised manuscript.
5. Conductivity equations in line 148 of previous manuscript have been deleted.

REVIEWERS' COMMENTS:

Reviewer #1 (Remarks to the Author):

I think that authors of the article improved the text. Therefore I believe that it can be published in the present form.

Reviewer #2 (Remarks to the Author):

I recommend publication of this paper in Nature Communications.

Reviewer #3 (Remarks to the Author):

This is my second review for this manuscript. As I stated in my first review, in large part, the importance of the present work is in correcting a prevalent misconception in the community. For this reason, attributing the negative longitudinal magnetoconductance in a non-topological material to disorder correctly questions previous claims that such observations were evidence for the chiral anomaly. The paper has several weaknesses pointed out in my previous report and by those of the other referees. However, I think the authors have done their best to seriously address all of the criticism. I am happy to support the current version for publication.

Responses to referee comments and recommendations

We are indebted to all three reviewers for their enthusiastic support in publishing our work in *Nature Communications*. Their insightful and constructive comments are well received and definitely helped to improve the quality of this manuscript. We revised our manuscript according to their recommendations and suggestions, with our point-to-point responses as follows.

Responses to comments and recommendations of Reviewer #1

Reviewer recommendation

The article reports an observation of negative longitudinal magnetoresistance in GaAs quantum wells. The observed magnetoresistances show dependences resembling those reported in topological materials. This is an important observation which call into question the conventional explanation of the phenomenon in the topological materials based of chiral anomaly. Therefore I believe that the article should be published.

Our response

We thank Reviewer 1 for highlighting the importance of our work and for recommending it to be published in *Nature Communications*.

Reviewer comment

My only advise to the authors is to make the theoretical part of the article more clear. In particular, the authors may clarify whether there is an explanation of the theoretical mechanism of the negative longitudinal magnetoresistance which they discuss in the article, or this is purely numerical results. This however should not delay the publication of the article.

Our response

We greatly appreciate Reviewer 1 for this advice, which was also echoed by Reviewer 3 as a weakness of the previous manuscript. In the revised manuscript we moved the numerical derivation of the various characteristic features of NLMR (Fig.3c-e of previous manuscript) to the supplement as Fig.S7 in the revised manuscript, with the caption detailing the analysis procedures and parameters.

As pointed out by Reviewer 3, we do not present a theory on the NLMRs. Instead, we propose a phenomenological three-resistor model to account for the various magnetic field behaviors of the NLMRs.

We significantly revised the text related to the three-resistor model in the manuscript.

Responses to Comments and Recommendations of Reviewer #2

Reviewer comment#1

Xu and co-authors report on the observation of a negative longitudinal magneto-resistance in GaAs quantum well. Their research is very interesting and timely. The experiments are carefully performed and the conclusions are solid.

The major issue I have with this paper is the connection and comparison of their data to results in existing literature concerning the observation of negative longitudinal magnetoresistance in topological

materials, in particular in Weyl semi-metals. The main difference can actually be seen just looking at figure 1e. As stated by the authors, negative longitudinal magnetoresistance (NLMR) is found above 150 K. In contrast, the NLMR in Weyl semi-metals is more pronounced at lower temperature and becomes weaker with increasing temperature, see e.g. Nature Communications, volume 7, Article number: 11615 (2016). Their simple resistor model is appropriate to describe their observations at high temperature with which they obtain very good fits, however, the underlying scattering mechanisms are related to the underlying systems, i.e. here, a quantum well with a finite well width. Specifically, the population of the second subband in the system seems to be crucial here. The authors mention that inter subband scattering is the dominant scattering mechanism in the temperature range in which NLMR is observed. I do not see how this is relevant, e.g. for Weyl semi-metals. It is actually not the same physics as in Weyl semi-metals but this type of scattering can also lead to NLMR based on entirely different mechanisms in completely different material systems.

I strongly recommend to tone down statements in the abstract and in parts of the introduction that make a direct connection between their 2D system and Weyl semi-metals. The material (systems) under study are completely different, however, a NLMR is observed which is supposed to have a completely different (microscopic) origin. To my opinion, the abstract does not represent the observations reported in the manuscript.

Our response

We are delighted that a leading expert says ‘their research is very interesting and timely. The experiments are carefully performed and the conclusions are solid.’

We followed the recommendation of Reviewer 2 to tone down the statements that make a direct connection between our 2D system and Weyl semi-metals. We re-wrote the abstract and re-phrased or removed statements related to Weyl semi-metals in the introduction.

We added the mentioned article [*Nat. Commun.* **7**, 11615 (2016)] to the references (Ref.18 in the revised manuscript)

Reviewer comment#2

Furthermore, I have one minor question and one minor recommendation.

- (1) I suggest adding self-consistent calculations for the system under investigation.*
- (2) Can the authors really refer to their sample as samples with ultra-high mobility?*
- (3) Is the expression ‘re-entrant temperature behavior’ justified in the abstract?*

Our response

- (1)** In the manuscript we present experimental observation and a phenomenological model. We hope that our model can stimulate more rigorous theoretical work, which is currently beyond our expertise.
- (2)** The mobilities (absolute values and temperature dependence) of our quantum wells are comparable to those in Ref.3 of this manuscript, which uses ‘ultra-high mobility’ to describe the quantum well in its title. In order to avoid confusion, however, we removed ‘ultra-high’ from the revised manuscript.
- (3)** We removed the term of ‘re-entrant’ in the revised manuscript.

Reviewer recommendation

To conclude, I emphasize that this manuscript is very relevant and addresses an important field in condensed matter physics, namely, the origin of the NLMR. I would recommend publication of this manuscript after taking into account my recommendations.

Our response

We thank Reviewer 2 for his/her positive judgement on our work and recommendation on publishing it in *Nature Communications*. As stated in our responses to his/her comments 1 and 2, we seriously took his/her insightful recommendations and revised the manuscript accordingly.

Responses to Comments and Recommendations of Reviewer #3

Reviewer recommendations

I would like to apologize to the authors for taking longer than usual for providing this report. I am generally supportive of the manuscript and think that a suitably revised version could appear in Nature Communications.

The experimental results are compelling, and would be a useful addition to the field. However, I find the theoretical analysis lacking and I also do not think the manuscript is well written. I encourage the authors to use the revision process to improve the presentation of results and the flow of ideas.

Our response

We thank Reviewer 3 for carefully reading the manuscript and for the constructive suggestions for improving the presentation of our results and the flow of ideas. We revised the manuscript according to his/her comment#2 below.

Reviewer comment#1

As mentioned above, the strongest part of the manuscript is the careful experimental data where the authors show the magnetoresistance for GaAs in a Hall bar geometry as a function of the magnetic field angle and temperature.

The authors correctly point out that a subset of the features they observe in this non-topological material, have in the previous literature been attributed to exotic topological effects including the chiral anomaly and the Berry curvature of the electronic bands. Perhaps, if these previous experimental papers had undergone more rigorous review, then such unfounded claims would not have entered the literature. However, given that the negative magnetoresistance is now commonly associated with the chiral anomaly, I am persuaded by the authors for the need for the present paper. This work will provide additional data to the community trying to understand magnetotransport in both topological and non-topological materials.

The other strength of the present manuscript is the discussion ruling out other explanations (e.g. the authors argue that weak localization quantum corrections used by previous analysis can explain some features of their data, but not others), etc. As I write below, this discussion should be expanded and done more clearly.

The weakest part of the paper is the analysis. The authors have a phenomenological equation (Eq 1 and 2) that they use to describe the experiments, plus a qualitative three-resistor model where they put in values by hand, and then get results consistent with Eq. 1 and Eq 2 (and the experimental observations). I would caution against even calling this a theory (as opposed to a qualitative picture). For example, a theory could involve solving the effective medium equations or solving a random resistor network model to capture the scenario shown in Fig. 4a with microscopic inputs, or a microscopic semi-classical Boltzmann calculation (all three of these have been done for the effect of disorder on the linear perpendicular magnetoresistance in other systems, but not for the negative parallel magnetoresistance).

So instead of a theory, the authors present a simple phenomenology that is consistent with their experiment. For the purpose of arguing that in principle, disorder can give rise to the kinds of features seen in the experiment, this is persuasive. The authors have persuaded me that disorder is likely to be the dominant effect, although the exact microscopic mechanism remains elusive. So while it is fine to say that impurities and surface roughness are consistent with the observations, without a theory for either, the language should be more cautious.

Our response

Yes, we present a simple phenomenological model rather than a theory to account for the observed NLMRs. We hope our work will stimulate more rigorous theoretical investigations and/or computer simulations on NLMRs in disordered systems. In this regard, we added the work by Parish and Littlewood [*Nature* **426**, 162 (2003)] to the references (Ref.42). In order to explain the non-saturating (linear) magnetoresistance for $B \perp I$, they modeled a strongly inhomogeneous conductor problem by discretization into a random resistor network that they analyzed numerically.

We agree with the reviewer on his/her judgement on the strengths and weakness of our previous manuscript. We revised the analysis part significantly (please see our response to the items #1-3 in his/her comments #2)

Reviewer comment#2

I also suggest the following to improve the manuscript:

- 1. I do not think it is feasible to ask the authors to do a full theory to explain their observations, although that would be nice. But at the very least they should write up exactly what they did (and what parameters they used) as a section of the supplemental material.*
- 2. Also, the longish discussion about a subset of some previous theories [27-30] that do NOT apply to the current experiments seems out of place in the flow of the paper. Maybe part of it could be moved into the supplementary material. Also, why discuss these theories and not others?*
- 3. At low temperatures weak localization corrections should become important. While I understand that this doesn't explain all the data in the current work, can the authors rule it out completely for the current experiment, especially at low temperature?*
- 4. Some acronyms like XMR are superfluous.*
- 5. Line 148, why write both resistivity and conductivity equations? Anybody can invert one to get the other.*

Our response

We greatly appreciate Reviewer 3 for these valuable suggestions on the presentation of this work, which definitely help to improve the quality of this manuscript. Following the reviewer's advices, we made the following changes:

1. We separated Fig.3 in the previous version into Fig.3 and Fig.S7 in the revised manuscript. We present a detailed explanation of the numerical calculations and their parameters in the caption of Fig.S7. The parameters derived using Eq.2 to fit the experimental data are presented in the caption of Fig.1 (for $T = 3$ K and 133K) and in Fig.4 (for $T \geq 138$ K).
2. We deleted the discussion (and the associated Fig.S7 in previous manuscript) about a subset of some previous theories [27-30], which were developed recently to explain the NLMRs without invoking chiral anomaly.

3. We expanded the discussion on possible explanation of NLMRs at low temperatures (data at $T = 3$ K) with weak localization or weak anti-localization corrections on pages 9 and 10 and a fitting curve with Eq.3 was added to Fig.S2b.
4. Superfluous acronyms XMR and 2DEG have been removed in the revised manuscript.
5. Conductivity equations in line 148 of previous manuscript have been deleted.